# Isolation and Identification of *Naegleria* Species in Irrigation Channels for Recreational Use in Mexicali Valley, Mexico

**DOI:** 10.3390/pathogens9100820

**Published:** 2020-10-07

**Authors:** Patricia Bonilla-Lemus, Saúl Rojas-Hernández, Elizabeth Ramírez-Flores, Diego A. Castillo-Ramírez, Alejandro Cruz Monsalvo-Reyes, Miguel A. Ramírez-Flores, Karla Barrón-Graciano, María Reyes-Batlle, Jacob Lorenzo-Morales, María Maricela Carrasco-Yépez

**Affiliations:** 1Laboratorio de Microbiología Ambiental, Grupo CyMA, UIICSE, FES Iztacala, UNAM. Av. De Los Barrios 1, Los Reyes Iztacala, Tlalnepantla de Baz 54090, Estado de Mexico, Mexico; blemus@unam.mx (P.B.-L.); erf@unam.mx (E.R.-F.); miigueram.96@gmail.com (M.A.R.-F.); karla-.ale@hotmail.com (K.B.-G.); 2Laboratorio de Inmunobiología Molecular y Celular, Sección de Estudios de Posgrado e Investigación. Escuela Superior de Medicina, Instituto Politécnico Nacional, Salvador Díaz Mirón esq. Plan de San Luis S/N, Miguel Hidalgo, Casco de Santo Tomas, Ciudad de Mexico 11340, CDMX, Mexico; srojash@ipn.mx (S.R.-H.); bono040388@gmail.com (D.A.C.-R.); 3Departamento de la Unidad de Secuenciación, Bioquímica Molecular, UBIPRO, FES Iztacala, UNAM. Av. De Los Barrios 1, Los Reyes Iztacala, Tlalnepantla de Baz 54090, Estado de Mexico, Mexico; reyesac2001@gmail.com; 4Instituto Universitario de Enfermedades Tropicales y Salud Pública de Canarias, Universidad de La Laguna, Avda. Astrofísico Fco. Sánchez, S/N, 38203 La Laguna, Tenerife, Islas Canarias, Spain; mreyesba@ull.edu.es (M.R.-B.); jmlorenz@ull.edu.es (J.L.-M.)

**Keywords:** *Naegleria* spp., *Naegleria fowleri*, identification, irrigation channels, Mexicali Valley

## Abstract

Members of the genus *Naegleria* are free-living amoebae that are widely distributed in water and soil environments. Moreover, *Naegleria fowleri* is a pathogenic amoeba species that causes a fatal disease in the central nervous system known as primary amoebic meningoencephalitis (PAM) in humans. Since most reported infections due to *N. fowleri* are reported in recreational waters worldwide, this study was aimed to describe the presence of these amoebic genus in Mexicali Valley irrigation channels of recreational use. A total of nine water samples were collected and processed by triplicate, in nine different sites of the Valley. After filtering and culturing the samples, plates were examined, and the observed amoebae were morphologically identified at the genus level. In addition, the pathogenicity of these amoebic isolates was checked, and molecular characterization was performed by PCR/sequencing. The results revealed the presence of *Naegleria* spp. in all the channels sampled. Finally, molecular identification confirmed the presence of five different species of *Naegleria: N. fowleri, N. australiensis, N. gruberi, N. clarki* and *N. pagei*. The presence of these protists, particularly *N. fowleri*, should be considered as a potential human health risk in the region.

## 1. Introduction

Free-living amoebae (FLA) are widely distributed protists in nature. Among them, some species are causative agents of a fatal brain infection in humans, including *Naegleria*, *Acanthamoeba, Balamuthia* and *Sappinia* [1,2,3]. *Naegleria fowleri* is a free-living thermophilic amoeboflagellate that causes primary amoebic meningoencephalitis (PAM) [1,2]. PAM is a rapidly spreading and fatal disease with a fatality rate of approximately 95% worldwide [4]. Infection of the brain occurs after amoebae reach the nasal cavity and invade the nasal mucosa. From there, amoebae penetrate the nasal epithelium and migrate by the olfactory nerves through the cribriform plate to invade the brain and meninges. Death usually occurs within ten to fifteen days of clinical manifestations appearing [1]. The disease is rare; however, new cases are being reported every year in the world [5,6]. 

Infection is mainly associated with the recreational use of water in swimming pools and water parks, naturally heated aquatic ecosystems, industrial refrigeration water and with the heating of lakes or ponds that electricity generating stations use as part of their cooling systems. There is a major health risk during warm weather seasons since these amoebae are thermophilic and their optimal proliferation temperature may reach 40 °C or even 45 °C. Therefore, the infection rate is higher and more probable during the summer season [2,7]. In the majority of cases, PAM occurs in young people who have recently been exposed or had contact with water containing *Naegleria fowleri* [1]. 

The first human case of PAM was described in Australia in 1965 [8]. So far, approximately 300 cases of PAM have been documented worldwide and only approximately 5% of the patients have survived [9]. Accordingly, PAM is considered a serious public health problem due to the previously mentioned high mortality rate [7,10,11]. Interestingly, a recent study showed that in Pakistan between 2008 and 2017, more than 100 cases were reported [12]. 

In Mexico, around 11 cases of PAM have been reported from 1984 to 2007 [12,13,14,15,16,17]. Unfortunately, there are many cases underdiagnosed around the world due to lack of awareness of the disease or the similarity of clinical symptoms with other more common central nervous systemdiseases such as bacterial meningitis and viral encephalitis [6,18]. 

The Mexicali Valley in Baja California (B.C) has an intensive agricultural system that depends on an irrigation system originating from the Colorado River. This irrigation system consists of a network of channels and reservoirs whose volumes vary with the irrigation cycle. The Valley is characterized by a desertic environment and high temperatures are prevalent, especially during the summer months. No recreational facilities are available in the valley, although swimming and wading in the irrigation channels are common practices in the area. Previous reports have shown a clear association of PAM with a history of swimming in these channels [14,19]. In 1993, Lares-Villa reported the presence of *Naegleria fowleri*, isolated in five fatal cases of PAM in Mexicali (B.C). Moreover, they also detected that the same species of amoeba was present in the irrigation channels and in the deceased patients [19]. It is important to mention that the population of the region knows about the disease that these free-living amoebae cause and it is not uncommon, mainly in summer, to see news of deadly of PAM cases in the local media. Unfortunately, they are not reported in the scientific literature [20].

Thus, this area was selected because most of the clinical cases in Mexico have been reported in this region [13,14,19]. Despite this, to date, there are a number of human settlements on the banks of irrigation channels and local inhabitants often swim in them in the summer when temperatures can be as high as 50 °C. Therefore, the aim of this study was to evaluate the presence of *Naegleria* species in Mexicali Valley irrigation channels, which despite warnings on the presence of *N. fowleri*, are still being used by the people as recreational aquatic sites.

## 2. Results

### 2.1. Physicochemical Parameters

A total of nine sites belonging to four different channels and three shared lands (also known as Ejidos in Mexico) were analyzed. Table 1 summarizes the obtained values related to the physicochemical water parameters. The lowest temperature was 16 °C, while the highest temperatures reached 20 °C. The observed conductivity varied from 1.4 × 10^3^ to 1.8 × 10^3^. The pH range was 7.3 to 7.7 and dissolved oxygen between 3.6 and 4.2 mg/L.

### 2.2. Morphological Description of Naegleria

Eleven *Naegleria*-like strains were isolated in all the sites included in this study. In some cases, more than one strain was isolated, as was the case for isolates CRES3, CRES4, and CRET (Table 2). The isolates shared common patterns of morphological characteristics [21], which are shown in Figure 1A–C. 

Trophozoite showed uroid, lobopod, and multiple vacuoles in the cytoplasm; measures of 20 to 35 µm in length with an average of 27.5 µm. At the anterior end, a wide hyaline lobopod was observed, and at the posterior end a prominent villous uroid was observed (Figure 1A). Cysts were round, smooth and of double wall with pores; they measured on average 15 µm in diameter; the nucleus shows chromatin granules in the periphery, and a centric dense small nucleolus surrounded by a clear halo (Figure 1B). Flagellate was observed with a pear shape, and usually with two flagella: they measured on average 13 µm in length (Figure 1C). All these observed morphological characteristics are compatible to the *Naegleria* genus.

### 2.3. Pathogenicity Test

After morphological identification of the obtained *Naegleria* isolates, pathogenicity testing of the eleven isolates was performed. Four isolates killed 100% of the mice within the first ten days (Table 3). To confirm that mice died due to PAM, amoeba were recovered from brain and lungs. 

### 2.4. Molecular Characterization 

Isolates morphologically identified as *Naegleria* were then subjected to PCR assay for their identification at the species level. The primers used successfully amplified the ITS (Internal transcribed spacer) with 5.8S rDNA of *Naegleria* isolates. The molecular sizes of the ITS amplicons were between 325 and 408bp (Figure 2, Table 4). 

### 2.5. Sequencing and Homology Analyses

The resulting PCR products were sequenced, edited, and compared with each other and with reference sequences. For comparative phylogenetic analysis, a search using BLAST of ITS for *Naegleria* sequences was performed. DNA sequencing analysis revealed the occurrence of some *Naegleria* species including *N. gruberi*, *N. australiensis*, *N. fowleri, N. clarki* and *N. pagei* (Table 4). The sequences of 11 amplicons of ITS-5.8S-ITS2, shown in Figure 2, correspond to the 11 isolates which were aligned using MEGA X. Interestingly, isolates CRES3a and CRES4b showed 100% homology with *N. fowleri*, meanwhile for isolates CRES2, CRES3b and CDENL, we found a 98, 98 and 97% homology with *N. australiensis*, respectively. In addition, isolates CRES1 and CRES4a showed 98 and 97% homological identities with *N. gruberi*, respectively. Isolates CRET1 and CRET2 showed 100% homological identities with *N. clarki* (Table 4). It is worth mentioning that especially isolate CRET1 was confirmed using another set of primers, which are also able to detect Vahlkampfiids (Vahl-1-FOR 5’-GTCTTCGTAGGTGAACCT-3’) and (Vahl-2-Rev 5’-CCGCTTACTGATATGCTTAA). With this set of primers, such an isolate was also identified as *N. clarki*.

Finally, isolates C4abrilENL and CRENL showed 100% homological identities with *N. pagei* (Table 4).

### 2.6. Phylogenetic Tree

Phylogenetic analysis was performed using Neighbor-Joining method, this test was conducted to assess phylogenetic relationships between each isolate and *Naegleria* sequences of reference species such as *Naegleria gruberi* (AJ132031.1 and AJ132031.1), *Naegleria australiensis* (AJ132034.1, AB128053.1 and AB128052.1), *Naegleria fowleri* (X96564.1 and KT375442.1), *Naegleria clarki* (AJ566625.1 and X96575.2) and *Naegleria pagei* (AJ566633.1 and JX267141.1). Similar sequences were grouped together (Figure 3). 

## 3. Discussion

*Naegleria* isolates were morphologically identified at the genera level by morphological criteria according to Page [21], in all sites analyzed (Table 2). Interestingly, four *Naegleria* isolates were able to kill 100% of infected mice used in their pathogenicity test; such isolation corresponded to CRES2, CRES3a, CRES3b and CRES4b (Table 3). Based on PCR, homology analysis of the obtained DNA sequences and phylogenetic analysis from these isolates, *N. fowleri* and *N. australiensis* were identified in the studied irrigation channels (Table 4). 

The most frequently isolated *Naegleria-*like amoeba belonged to the *N. australiensis species*. Interestingly, this species has already shown pathogenicity in animal models [3,22]. In the present work, CRES2 and CRES3b isolates were able to grow at 30, 37 and 42 °C, indicating their potential pathogenicity, which was confirmed after all infected mice died. Only the CDENL strain also identified as *N. australiensis* did not kill mice, probably due to its long time in cultivation. However, trophozoites were found in the brain, lung, kidney, and liver, after 21 days, when mice were euthanized. Hence, the isolation and identification of *N. australiensis* in the studied aquatic ecosystems should be an important issue to be explored in further studies due to its capability to cause encephalitis in experimental animals. Moreover, its presence in the channels of Mexicali could also represent a public health threat along with *N. fowleri.*

*N. fowleri* was identified in 1993 from that area by growth characteristics, serological study and isoenzimes patterns [19]. Therefore, this is the first morphological and molecular study of amoebas of the genus *Naegleria* isolated from the water of the irrigation channels of the Mexicali Valley that describes the presence not only of the pathogenic *N. fowleri* and *N. australiensis* species, but also of non-pathogenic species such as *N. gruberi*, *N. clarki* and *N. pagei* (Table 4). 

However, the present study highlights its importance in the detection and identification of *N. fowleri* since it is the only pathogenic species for humans. 

The Mexicali Valley is characterized by a desertic environment and high temperatures, especially during the summer months where air temperatures can reach as high as 50 °C. However, *N. fowleri* strains were obtained in the autumn season (November) with water temperatures of 16 °C and 17 °C (Table 1, CRES4b and CRES3a, respectively). 

*N. fowleri* has been found almost exclusively during warmer months when the temperatures are above of 30 °C [3], but this is not a rule. In a study performed by Sifuentes et al. (2014) in recreational waters in Arizona, *N. fowleri* was found in winter and spring seasons and, less frequently, also in autumn [23]. Another study in Oklahoma related to the presence of *N. fowleri* in surface waters where five *N. fowleri* isolates were found in the summer (29.2 °C) and one in autumn (12 °C) [24]. Whereas in a study of surface waters in Connecticut conducted during the summer, no correlations were observed between the presence of *N. fowleri* and the temperature, which was above 20 °C [25]. Other studies have suggested that water temperature is not an important factor affecting the distribution of *N. fowleri* since these amoebas have been isolated from water with temperatures ranging from 16 °C to 45 °C [26,27,28].

In the present work, the dissolved oxygen was between 3.6 and 4.2, water electrical conductivity from 1400 to 1800 µS /cm and the pH of water samples ranged from 7.3 to 7.7 (Table 1). These values were not restrictive for the growth of the species of *Naegleria* identified. Nevertheless, it is known that *N. fowleri* prefers a lower pH, growing best at a slightly acidic pH of 5.5 to 6.5 [29]. The results showed that *N. fowleri* is probably being inhibited since only two isolates from eleven of the pathogenic amoebae (CRES3a and CRES4b) were obtained (Table 4). This fact might also be related to the water temperature of channels, which was between 16 and 20 °C and, as it was mentioned above, it is not the ideal environment for *N. fowleri*. All these results lead us to conclude that there could be multiple environmental factors that affect the occurrence of *N. fowleri* in surface waters in addition to salinity and pH, as suggested by a previous report [30]. 

To select pathogenic strains, it is important to test their ability to proliferate at 42 °C. However, not only pathogenic strains are thermophilic, and some others of the amoebae can proliferate at this temperature or higher [31,32]. Thus, the temperature tolerance test is not enough to determine the pathogenicity of the amoebae and for this reason we use the pathogenicity test to know its ability to cause PAM, which resulted positive, as was reported above. Moreover, the presence of *N. fowleri* and the other isolates were confirmed by the sequencing of the ITS regions and phylogenetic analysis (Figure 3). 

It is known that ribosomal DNA (rDNA) is a powerful tool to study phylogenetic relationships of organisms at different taxonomic levels. However, 18s rDNA fragment analyses have shown very few differences between *N. fowleri* and the closely related *Naegleria lovaniensis* species, indicating that this gene region is not sufficiently variable for intraspecific studies. On the other hand, ITS regions, which are subject to lower functional constraints than the rDNA genes, have evolved more rapidly and, thus, these regions are more appropriate to detect differences between and within species. ITS analysis has been used successfully at the specific and intraspecific level for the phylogenetic analysis and typing of various organisms.

In general, the role of ITS is to participate in the regulation and processing of the genes that code for the small subunit (18 s) and the large region of ribosomes, and, once the polycistronic rRNA is synthesized, the ITS regions are eliminated. These regions are highly variable, so a genetic signature can be generated for each species, which allows it to be used to identify different species of the same genus, in this case for the *Naegleria* species. It is worth mentioning that, apparently, the ITS regions only participate in the regulation and synthesis of ribosomal RNA (rRNA), since they are eliminated once the polycistronic rRNA is formed and, therefore, these regions (ITS) are not transcribed [33]. In this way, these regions do not participate in protein expression (virulence factor). In a future study, it would be interesting to perform a screening of the major virulence factors (protein level) among the eleven isolates. For example, in a recently published work by our group, we identified differential proteins and protein pattern recognition between *Naegleria fowleri* and *Naegleria lovaniensis* using anti-*N fowleri* antibodies as a strategy to find vaccine candidates against meningoencephalitis. The results obtained showed very notable differences in spot intensity between these two species by 2-DE gels and 2-DE Western blot, specifically those with relative molecular weights of 100, 75, 50 and 19 kDa. Some spots corresponding to these molecular weights were identified as actin fragment, myosin II, heat shock protein, and membrane protein Mp2CL5, among others, with differences in theoretical post-translational modifications [34]. Interestingly, it has been shown that some pathogenicity factors are not exclusive to *N fowleri*, since they have also been reported in nonpathogenic *Naegleria* species with different levels of expression [35,36,37,38]. Therefore, in the future, we could perform a similar study with the different species found in the present work.

Since no recreational facilities are available in the valley, swimming and wading in the channels are pastimes for the people who live in the valley. These channels may form stagnant ditches that may be contaminated by free living amoebae. Because the water of these channels is running water, it is not viable to chlorinate the water of the channels [19]. Moreover, the water of these channels is used by the local people as their main domestic source where they tend to chlorinate their water for disinfection, but do not receive training on the most effective way to use chlorine.

The reported human cases of PAM in this area have been during the summer season when the environmental temperature reaches 50 °C, leading to increased water temperature of irrigation channels [13,14,19], which facilitate the proliferation of pathogenic amoebae when there is a confluence or large number of local people and visitors swimming in these channels. 

It is worth mentioning that PAM is difficult to diagnose because the clinical signs of the disease are similar to bacterial meningitis [1]. In Mexico, few data about detection of *N. fowleri* in aquatic environments are available, thus the findings of the present study would help to prevent recreational activities in places where *N. fowleri* is found. We need to clearly develop an effective amebae-monitoring program that includes rapid, specific and sensitive assays which consider physical, chemical, and biological factors associated with the presence and dynamics of *N. fowleri* in the environmental systems in endemic areas such as Mexicali Valley. Furthermore, there is an urgent need for increased awareness of the residents of the surrounding areas through some simple practices such as avoiding water entering the nasal passages. This includes keeping the head above water or using nose clips when swimming. It is also important to avoid recreational activities in freshwater during periods of high water temperatures. 

## 4. Conclusions

Thermophilic *N. fowleri* and *N. australiensis* were present in one of the coldest months of the year in the channels of Mexicali Valley. In this study, *N. gruberi*, *N. pagei* and *N. clarki* were also identified, species that had not been previously described for this place and which represent new knowledge regarding the diversity of *Naegleria* in that area of Mexico. Rising temperatures in recent years due to global warming, together with poor infrastructure of wastewater management and sanitation as well as drug resistance, will cause a further rise in the number of deaths due to infectious diseases, and this is not an exception for PAM. Furthermore, this disease goes unnoticed or is not reported as the symptoms can be confused with other infections of the central nervous system, and consequently, the number of cases can be underestimated. There is an urgent need to increase the awareness of the public as well as health professionals, and for a mixture of educational and behavioral modification strategies, in order to avoid serious consequences for people. It is also necessary to develop rapid methods of early detection of the amoeba in environmental and clinical samples that are consistent with the resources and infrastructure of each country, as is the case for Mexico.

## 5. Materials and Methods

### 5.1. Sampling

In total, 9 water samples were collected in November 2016 from irrigation channels of three different shared lands or ejidos from the Mexicali Valley (Figure 4). A 250-mL sample of water was taken on each site by triplicate. Water samples were placed in sterile containers and kept at room temperature until analysis. Water temperature, conductivity (K_25_) and pH were measured in situ using a conductimeter model PC18 (Conductronic Instruments, Puebla, Mexico); dissolved oxygen was measured with an oxymeter YSI Model 51 B. These analyses are adapted from standard methods for the examination of water and wastewater [39]. All measurements were made in triplicate.

### 5.2. Processing Samples

Water samples were homogenized, and 100 mL were filtered through a 5.0 µm-diameter Millipore filter (Millipore, Bedford, MA, USA). Membranes were placed in Petri dishes with a non-nutrient agar medium with *Enterobacter aerogenes* ATCC-13048 (NNA) [21]. This procedure was carried out in duplicate to incubate at 30 °C and 37 °C. 

After 24 h of incubation, and up to seven days, NNA plates were examined through an inverted microscope (Zeiss model 473028) to detect growth of free-living amoebae. 

Amoebae were sub-cultivated in a fresh NNA medium several times to clean them and separate them from other microorganisms (bacteria, fungus etc.). Blocks (1 cm^2^) were taken from the NNA medium containing trophic amoebae and transferred to axenic culture in modified Chang’s liquid medium [40] and 2% (*w*/*v*) Bacto™ Casitone medium [41] (Difco, Le Pont de Claix, France) supplemented with 10% (*v*/*v*) fetal bovine serum (Gibco, Grand Island, NY, USA) and 100 U penicillin and 100 µg of streptomycin per milliliter. Cultures were incubated again at 30 °C or 37 °C to obtain pure cultures of the amoebae isolated [40].

### 5.3. Morphological Identification

In vivo preparations of the amoebae from axenic and/or monoxenic cultures were examined through a phase-contrast optical microscope at magnifications of 40× and 100× (Zeiss K7). The identification of *Naegleria* isolates was based on the morphological features of trophozoite and cyst, temperature tolerance and the flagellate transformation test [21].

### 5.4. Pathogenicity Test

All procedures performed in this study that involved Balb/c mice were in accordance with the Mexican federal regulations for animal experimentation and care (NOM-062-ZOO-1999, Ministry of Agriculture, Mexico City, Mexico) and approved by the ethical standards of the Facultad de Estudios Superiores Iztacala, UNAM (Number of Approval CE/FESI/022020/1317). In all experiments, 6 to 8-weeks-old male BALB/c mice were used. These mice were maintained 5 to 6 per cage and provided food and water ad libitum.

An important test to determine if a *Naegleria* strain is pathogenic is its capacity to kill mice; therefore, *Naegleria* isolates were evaluated by a pathogenicity test. This test was carried out on mice using axenic amoeba cultures [42,43]. Trophozoites were concentrated at 3000 rpm for ten minutes and were adjusted to a concentration of 1 × 10^5^ per ml. A volume of 0.02 ml was taken and was administered by intranasal inoculation. Five mice were inoculated with an amoeba-free culture medium for control purposes. The mortality rate was determined after the infected animals were monitored for up to 21 days or they were euthanized when they become moribund. Their brain, liver, lungs, and kidneys were removed and placed on non-nutrient agar medium that was incubated at isolation temperature (30 or 37 °C). Cultures were observed every day for a week to monitor the development of amoebae.

### 5.5. DNA isolation and PCR

Amoebae were cultivated axenically in a modified Chang’s medium or Bacto™ Casitone medium in 50 mL culture flasks and were incubated at isolation temperature. Cultures were harvested and centrifuged at 3500 rpm for 15 min and excess was disposed. From those thermophilic isolates positive to the flagellation and morphologically identified as *Naegleria* genus, DNA was extracted. Trophozoites (1 × 10^6^) were suspended in phosphate-buffer saline (PBS 1X) and a Zymo Reasearch Quick-gDNA™ MiniPrep kit (Zymo Research Corp., Irvine, California, United States of America, Catalogue # D3021) was then used for DNA extraction according to the manufacturer’s recommendations.

Polymerase Chain Reaction (PCR) was conducted according to the manufacturer’s instructions using a KAPA Taq ReadyMix PCR Kit (KAPA Biosystems, Catalogue # KK1006). Once the mix had been prepared, samples were placed in a Quanta Biotech S-24 thermal cycler and run under the following conditions: an initial 5 min denaturation at 95 °C followed by 39 cycles of amplification with a 1 min denaturing at 95 °C, a 2 min annealing at 58 °C, a 1 min extension at 72 °C and a final extension of 10 min at 72 °C. Since internal transcribed spacers (ITS) sequences have been used as a tool to resolve phylogenetic relationships between closely related *Naegleria* species, we have decided to use the following set of primers: The forward primer ITS-FOR 5’-GGGATCCGTTTCCGTAGGTGAACCTGC-3’ and reverse ITS-REV 5’-GGGATCCATATGCTTAAGTTCAGCGGGT-3’. These primers have been used for the selective PCR amplification of the whole ITS region (ITS1, 5.8S, and ITS2) [33,44,45]. *N. fowleri* (ATCC 30808) and *N. gruberi* (ATCC 30540) cultures were used as the positive control for the standardization of the PCR assay.

Amplicons (5 μL aliquot of the PCR reaction) were analyzed by electrophoresis on 2% agarose gel (SIGMA, St. Louis, MO, USA) in TBE buffer, stained with Midori Green Advanced (2.5 μL). The gel was then displayed on a UV transilluminator at a wavelength of 260 nm. 

### 5.6. Sequencing and Phylogenetic Analysis 

The PCR products of isolates were then used for sequencing in a HITACHI ABI PRISM^®^ 3100 Genetic Analyzer at the Biochemistry Molecular Laboratory of the Biotechnology and Prototypes Unit (UBIPRO) at the Facultad de Estudios Superiores Iztacala, UNAM. Sequencing data were analyzed and edited using the BioEdit software version 7.2.5, using FWD and RV sequence electropherograms as the base to edit any possible errors or artefacts and obtain a consensus sequence from both sequences, and to have a more reliable and accurate sequence.

For molecular evolutionary genetics analysis, MEGA X software was used for the multiple alignment of sequences [46] and to compare the sequences obtained with the sequences reported on GenBank using a Basic Local Alignment Search Tool (BLAST). Phylogenetic analysis was carried out based on the internal transcribed spacers (ITS) and 5.8S sequences using the same software. The *Naegleria* spp. identification was based on sequence homology analysis by comparison to the available ITS sequences in the GenBank database. Phylogenetic trees were constructed using the Neighbor-Joining tools [47] with a Bootstrap value of 1000 and a cut-off point of 50% [48]. The evolutionary distances were computed using the Maximum Composite Likelihood method [49]. Finally, the nucleotide sequences were submitted to the Genbank and the assigned accession numbers are shown in Table 4.

## Figures and Tables

**Figure 1 pathogens-09-00820-f001:**
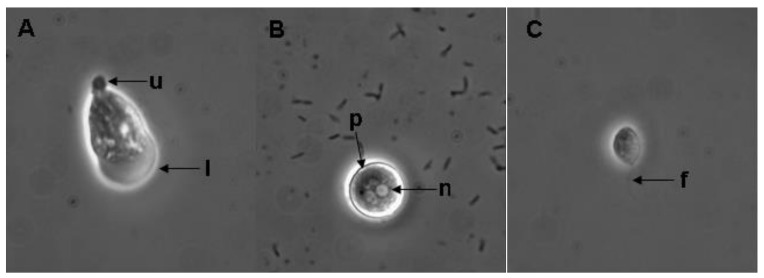
Morphological characteristics of CRES3b isolate of *Naegleria*: trophozoite (**A**), cyst (**B**) and flagellate (**C**). Lobopod (l), uroid (u), pore (p), nucleus (n), flagella (f). Magnification 400×. Phase contrast microscopy.

**Figure 2 pathogens-09-00820-f002:**
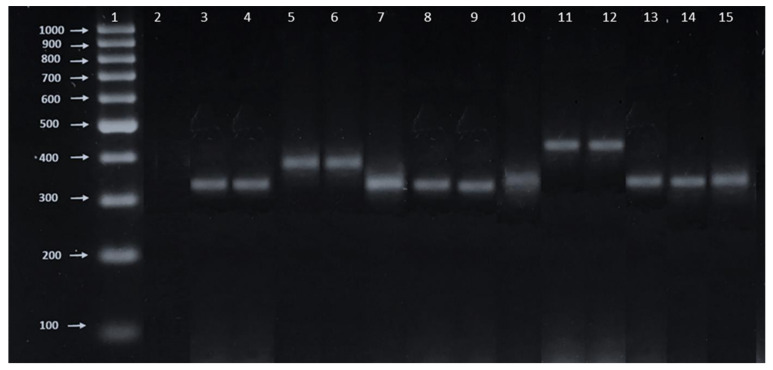
Obtained PCR amplicons in this study. Midori Green stained agarose gel showing the size variation of the ITS (ITS1, 5.8S rDNA and ITS2) PCR product obtained from *Naegleria* isolates. Lane 1: 100 bp DNA ladder. Lane 2: PCR reaction without DNA template as negative control. Lane 3: *N. fowleri* ATCC 30808 and Lane 4 *N. gruberi* ATCC 30540 as positive controls. Lane 5 to Lane 15 correspond to the isolates: 5: C4abrilENL; 6: CRENL; 7: CRES1; 8: CRES4a; 9: CRES3a; 10: CRES4b; 11: CRET1; 12: CRET2; 13: CRES2; 14: CRES3b and 15: CDENL. DNA amplicons separated by 2.5% agarose gel electrophoresis.

**Figure 3 pathogens-09-00820-f003:**
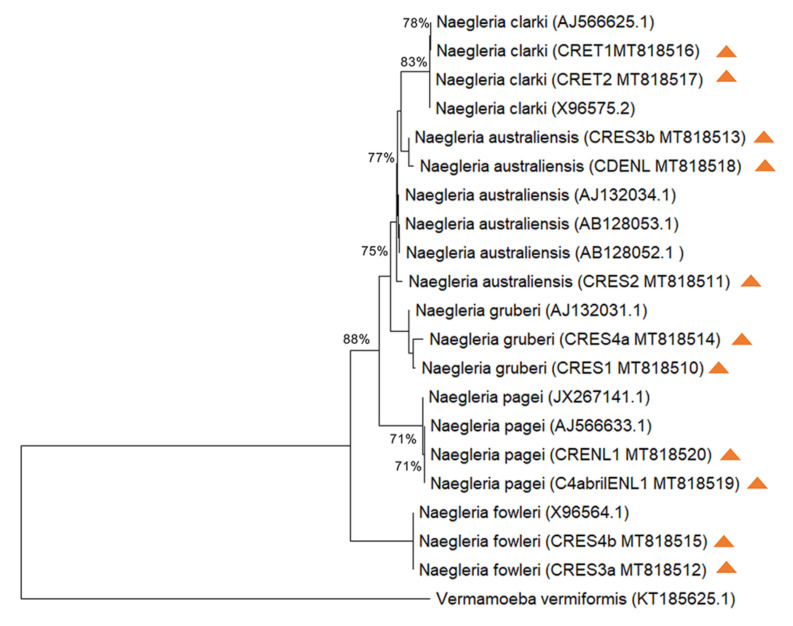
Neighbor-Joining tree depicting the relationships between the isolates with reference strains of *Naegleria* for ITS1, 5.85 and ITS2 sequences. The numbers on the tree branches are the percentages of bootstrap values. Bootstrap values (1000 replicates) are indicated for the nodes gaining more than 50% support. Symbols of different colors represent the isolates. *Vermamoeba vermiformis* was used as the outgroup. Orange triangles represent each isolate with their GenBank accession number.

**Figure 4 pathogens-09-00820-f004:**
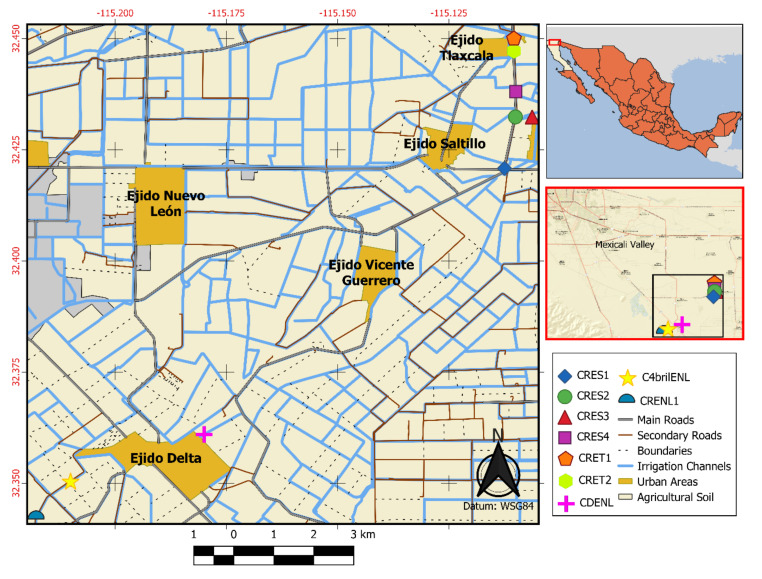
Map representing Mexicali Valley located in Northern Mexico that shows the irrigation channels (blue lines). Colored symbols are shown as the sampling sites (studied area).

**Table 1 pathogens-09-00820-t001:** Physicochemical parameters in the sites studied.

Sampling Area (CHANNELS AND SHARED LANDS)	Sampling Sites ID	pH	Water temperature °C	Dissolved Oxygen (mg/L)	Conductivity µS/cm
Canal Revolución Ejido Saltillo (CRES)	CRES1	7.5	17	4	1.6 × 10^3^
	CRES2	7.6	17	3.8	1.4 × 10^3^
	CRES3	7.5	17	3.6	1.5 × 10^3^
	CRES4	7.5	16	4	1.6 × 10^3^
Canal Revolución Ejido Tlaxcala (CRET)	CRET1	7.3	19	4.2	1.8 × 10^3^
	CRET2	7.6	17	4	1.4 × 10^3^
Canal Delta Ejido Nuevo León (CDENL)	CDENL	7.5	17	4	1.6 × 10^3^
Canal 4ABRIL Ejido Nuevo León (C4ABRILENL)	C4abrilENL	7.5	17	4.2	1.5 × 10^3^
Canal Reforma Ejido Nuevo León (CRENL)	CRENL	7.7	20	4.2	1.5 × 10^3^

CRES1–4: Canal Revolución Ejido Saltillo; CRET1–2: Canal Revolución Ejido Tlaxcala; CDENL: Canal Delta Ejido Nuevo León; C4ABRILENL: Canal 4 de abril Ejido Nuevo León, CRENL: Canal Reforma Ejido Nuevo León.

**Table 2 pathogens-09-00820-t002:** Characteristics of isolates of *Naegleria* from Mexicali Valley.

Sampling Area (CHANNELS SHARED LANDS)	Sampling Sites ID	Temperature Test (°C)	Flagellation Test	Positive *Naegleria* */ ID Isolates
Canal Revolución Ejido Saltillo (CRES)	CRES1	30 **, 37	+	+/CRES1
	CRES2	30, 37 **, 42	+	+/CRES2
	CRES3	30, 37 **, 42, 45	+	+/CRES3a
		30, 37 **, 42	+	+/CRES3b
	CRES4	30 **, 37	+	+/CRES4a
		30, 37 **, 42, 45	+	+/CRES4b
Canal Revolución Ejido Tlaxcala (CRET)	CRET1	30 **, 37	+	+/CRET1
	CRET2	30 **, 37	+	+/CRET2
Canal Delta Ejido Nuevo León (CDENL)	CDENL	30 **, 37, 42	+	+/CDENL
Canal 4ABRIL Ejido Nuevo León (C4ABRILENL)	C4abrilENL	30 **, 37	+	+/C4abrilENL
Canal Reforma Ejido Nuevo León (CRENL)	CRENL	30 **, 37	+	+/CRENL1

* By morphological identification. ** Optimal temperature.

**Table 3 pathogens-09-00820-t003:** Pathogenicity test.

Strain	Species	% Mortality	Mice Death (in Days)	Amoeba Recovered From:
CRES1	*Naegleria gruberi*	0	-	-
CRES2	*Naegleria australiensis*	100	5–10	Brain and lung
CRES3a	*Naegleria fowleri*	100	7–8	Brain
CRES3b	*Naegleria australiensis*	100	5–7	Brain and lung
CRES4a	*Naegleria gruberi*	0	-	-
CRES4b	*Naegleria fowleri*	100	7–8	Brain
CRET1	*Naegleria clarki*	0	-	-
CRET2	*Naegleria clarki*	0	-	-
CDENL	*Naegleria australiensis*	0	-	-
C4abrilENL	*Naegleria pagei*	0	-	-
CRENL1	*Naegleria pagei*	0	-	-

**Table 4 pathogens-09-00820-t004:** Result of DNA sequencing of the selected amplicons.

Isolate/ Amplicons Length (bp)	Assigned Accession No.	Closest Phylogenetic Species	Reference Strain Accession No.	Coverage Percentage with Reference Strain
CRES1/325	MT818510	*Naegleria gruberi*	AJ132031.1	98%
CRES2/311	MT818511	*Naegleria australiensis*	AJ132034.1	98%
CRES3a/352	MT818512	*Naegleria fowleri*	X96564.1	100%
CRES3b/325	MT818513	*Naegleria australiensis*	AB128053.1	98%
CRES4a/325	MT818514	*Naegleria gruberi*	AJ132031.1	97%
CRES4b/352	MT818515	*Naegleria fowleri*	KT375442.1	100%
CRET1/408	MT818516	*Naegleria clarki*	AJ566625.1	100%
CRET2/437	MT818517	*Naegleria clarki*	X96575.2	100%
CDENL/325	MT818518	*Naegleria australiensis*	AB128052.1	97%
C4abrilENL/373	MT818519	*Naegleria pagei*	AJ566633.1	100%
CRENL/373	MT818520	*Naegleria pagei*	JX267141.1	100%

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
