# Peer review of "Isolation and Identification of *Naegleria* Species in Irrigation Channels for Recreational Use in Mexicali Valley, Mexico"

_pathogens, 2020, doi:10.3390/pathogens9100820_

Round 1
Reviewer 1 Report
This article brings new knowledge regarding diversity of Naegleria in that area of Mexico. The study showed that is important to monitor water quality and that it is necessary to increase awareness of the public and health professional. However, English needs improvement and punctuation needs to be checked. In addition, the authors should add more detail in certain passages of the article so that readers can have a better understanding. For a better quality of the article, the authors must make corrections and answer point by point to each request of the reviewer.

Author Response
REVIEWER 1
Comments and Suggestions for Authors
This article brings new knowledge regarding diversity of Naegleria in that area of Mexico. The study showed that is important to monitor water quality and that it is necessary to increase awareness of the public and health professional. However, English needs improvement and punctuation needs to be checked. In addition, the authors should add more detail in certain passages of the article so that readers can have a better understanding. For a better quality of the article, the authors must make corrections and answer point by point to each request of the reviewer.
Abstract:
- Line 34: Modify the sentence please by ”The presence of these protists”
- It was done in line 33
Introduction:
- Line 39: Modify the sentence please by “Free-living amoebae (FLA) are widely distributed
protists in nature”
- It was done in line 38
- In line 39 to 41: You need to add references for each of these sentences please.
- It was done in lines 40, 41 and 42
- In line 38: You need to remove the capital letters “primary amoebic meningoencephalitis
(PAM)”
- It was done in lines 40 and 41
- In line 64: Modify the sentence by: “The Mexicali Valley in Baja California (B.C)”.
- It was done in line 63
- In line 71: Modify the sentence by:
“In 1993, Lares-Villa reported the presence of Naegleria fowleri, isolated in five fatal cases of
PAM in Mexicali (B.C).”
- It was done in lines 69 and 70
- In line 76:
You need to add a point at the end of the line
“Unfortunately, they are not reported in the scientific literature [18].”
- it was done in line 75
Results:
- In line 88: You forgot the period at the end of the sentence, modified the sentence please:
“The pH range was 7.3 to 7.7 and dissolved oxygen between 3.6 and 4.2.”
- It was modified, and the period was added. Lines 87 and 88
- Line 178: In table 3, you need to add a column with of the coverage (in percentage) of each
amplicon that were compared with nucleotides contained in public database.
- It was done, now is as “Coverage Percentage with Reference Stain” now Table 4
- Line 181: Figure 3.
For the phylogenetic tree, if possible, you need to add outgroup (for example: organisms
belonging to Acanthamoeba sp., Vermamoeba sp., Vahlkampfiaa sp. or Willaertia sp.). You
can search in NCBI (genbank) the sequences of ITS1-5,8S-ITS2 of organisms
phylogenetically close to Naegleria species.
- In fig. 3, an outgroup was added (Vermamoeba vermiformis)
You must let appear only the percentage higher than 70% on the figure.
- Now, only the percentage higher than 70% appears
In the legend, it should be specified with which color is associated each triangle, please.
- It has been done. Only orange triangles were used to specific the isolates as well as its GenBank accession number was added
Discussion
- Line 224 : Modify the sentence by “in the summer (29.2°C) and one in autumn (12°C) [23]”
- It was done in line 225
Material and methods:
- Line 276: You need to add a reference so that readers check what is the Conductronic Mod, please.
«situ using a Conductronic Mod »
- It was done in line 291
- Line 276: What is PC18?
- It is the model. It was corrected in line 291
- Line 277: Capital letters on these words are unnecessary, please remove them.
“Standard Methods for the Examination of Water and Wastewater [32].”
- It was done in lines 292 and 293
- Line 286: You must specify the Enterobacter aerogenes strain used for the isolation of the amoeba please.
- It was done in line 301
- Line 289: You need to define the name and the model of inverted microscope used for the detection of the amoebas.
- It was done in line 304
- Line 291 to 295: Initially the amoebae were isolated on xenic cultures from non-nutritive agar in the presence of bacteria. Subsequently, you have cultivated the amoebae on an axenic rich medium. You do not specify if you used an antibiotic or antifungal cocktail to grow the amoeba in axenic environments. Please describe the transition from xenic to axenic growth more precisely.
- It was done in lines 305 to 311
- Line 293: You must describe the composition of the media (Chang and BC) or add references that described the composition of media, please.
- It was done in line 307 and 308
- Line 355: You need to add a reference for this sentence please.
“evolutionary distances were computed using the Maximum Composite Likelihood method”
- It was added in line 371
- Line 356: Conclusion
The conclusion is not in the right place. The conclusion must be found after the discussion,
you need to change the place of the conclusion.
- It was done, now the conclusions are in lines from 270 to 283
- Line 369: You forget the period at the end of the sentence, modified the sentence please:
“infrastructure of each country as is the case for Mexico.
- The period was added at the end of the conclusion, line 283
Submission Date
31 July 2020
Date of this review
13 Aug 2020 00:10:25

Reviewer 2 Report
An interesting study focused on Naegleria species with important environmental impact. Yet, the manuscript is more technical and some scientific and biological pieces of knowledge are missing to make the study more comprehensive. Please find below my comments:
>>> Pathogenicity test: Authors should show the survival curves of the mouse experiments and the amoeba count from brain, lungs and kindey. In the current version, the data shown in the Table 1 are not very comprehensive.
>>> Molecular characterization: What does the ITS expression tell of the protein expression? It would be important that authors perform a screening of the major virulence factors (protein level) among the 9 water samples collected.
>>> General comment: Can authors speculate whether some novel virulence factors could be involved in the 9 water samples collected?
Author Response
REVIEWER 2
Comments and Suggestions for Authors
An interesting study focused on Naegleria species with important environmental impact. Yet, the manuscript is more technical and some scientific and biological pieces of knowledge are missing to make the study more comprehensive. Please find below my comments:
- >>> Pathogenicity test: Authors should show the survival curves of the mouse experiments and the amoeba count from brain, lungs and kindey. In the current version, the data shown in the Table 1 are not very comprehensive.
- A new table is shown with the mortality data (Table 3). We do not have the amoeba count since our interest in this study was to know if Naegleria isolates are pathogenic in mice and if they are capable of invading other organs. However, the count will be considered in future studies.
- >>> Molecular characterization: What does the ITS expression tell of the protein expression?
- Ribosomal DNA (rDNA) has proven to be a powerful tool to study phylogenetic relationships of organisms at different taxonomic levels. However, 18s rDNA fragment analyzes have
shown very few differences between N. fowleri and the closely related Naegleria lovaniensis species indicating that this gene region was not sufficiently variable for intraspecific studies.
Ribosomal internal transcribed spacers (ITS), which are subject to lower functional constraints than the rDNA genes, have evolved more rapidly and, therefore, are more suitable for detecting differences between and within species. ITS analysis has been used successfully at the specific and intraspecific level for phylogenetic analysis and typing of various organisms
In general, the role of ITS is to participate in the regulation and processing of the genes that code for the small subunit (18s) and the large region of ribosomes, once the polycistronic rRNA is synthesized, the ITS regions are eliminated. These regions are highly variable, so a genetic signature can be generated for each species, which allows it to be used to identify different species of the same genus, in this case, Naegleria.
Apparently, the ITS regions only participate in the regulation and synthesis of rRNAs, since they are eliminated once the polycistronic rRNA is formed and therefore these regions (ITS) are not transcribed. In this way, these regions do not participate in protein expression (virulence factor).
It would be important that authors perform a screening of the major virulence factors (protein level) among the 9 water samples collected.
- This would be very important in future studies. For example, in a recently published work by our group, we identified differential proteins and protein pattern recognition between Naegleria fowleri and Naegleria lovaniensis using antibodies anti-N fowleri as strategy to find vaccine candidates against meningoencephalitis. The results obtained showed very notable differences in spot intensity between these two species by 2-DE gels and Western blot, specifically those with relative molecular weight of 100, 75, 50 and 19 kDa. Some spots corresponding to these molecular weights were identified as actin fragment, myosin II, heat shock protein, membrane protein Mp2CL5 among others, with differences in theoretical post-translational modifications.
- Gutierrez-Sanchez, M.; Carrasco-Yepez, M.M.; Herrera-Diaz, J.; Rojas-Hernandez, S. Identification of differential protein recognition pattern between Naegleria fowleri and Naegleria lovaniensis. Parasite Immunol. 2020, 42, e12715.
In this way, in a future, we could perform a similar study with the different species found in the present work. It is worth mentioning that some pathogenicity factors are not exclusive to N fowleri, since they have also been reported in nonpathogenic Naegleria species with different level of expression.
- Marciano-Cabral F, Cabral GA. The immune response to Naegleria fowleri amebae and pathogenesis of infection. FEMS Immunol Med Microbiol. 2007;51(2):243-259.
- Jamerson M, da Rocha-Azevedo B, Cabral GA, Marciano-Cabral F. Pathogenic Naegleria fowleri and non-pathogenic Naegleria lovaniensis exhibit differential adhesion to, and invasion of, extracellular matrix proteins. Microbiology. 2012;158(Pt 3):791-803.
- Cursons RT, Keys EA, Brown TJ, Learmonth J, Campbell C, Metcalf P. IgA and primary amoebic meningoencephalitis. Lancet. 1979;1(8109):223-224.
- Cervantes-Sandoval I, Jesus Serrano-Luna J, Pacheco-Yepez J, Silva-Olivares A, Tsutsumi V, Shibayama M. Differences between Naegleria fowleri and Naegleria gruberi in expression of mannose and fucose glycoconjugates. Parasitol Res. 2010;106(3):695-701.
- Carrasco-Yepez M, Campos-Rodriguez R, Godinez-Victoria M, et al. Naegleria fowleri glycoconjugates with residues of alpha-D-mannose are involved in adherence of trophozoites to mouse nasal mucosa. Parasitol Res. 2013;112(10):3615-3625.
- >>> General comment: Can authors speculate whether some novel virulence factors could be involved in the 9 water samples collected?
- According to the previously mentioned, it is likely by using tools such as 2-DE and 1-DE Western blot and mass spectrometry in order to search virulence factors from these samples collected.
Submission Date
31 July 2020
Date of this review
24 Aug 2020 12:21:48

Reviewer 3 Report
The manuscript "Isolation and identification of Naegleria species in irrigation channels for recreational use in Mexicali Valley, Mexico" reports the presence of Naegleria in different areas in Mexicali Valley. This work is of interest in the field, as there are few reports about Naegleria infections and its incidence is understimated. Research design is appropriate and results are presented clearly. However, some changes must be done before being accepted to be published in Pathogens.
- Manuscript elaboration. There are some grammar and punctuation mistakes that must be revised (for example: line 23, line 33, line 34, line 47, line 240, line 308: “…,it is capacity…”, etc.). Besides, some sentences may be rephrased to make them easier to understand. For example, lines 41-42, lines 44-47 (sentence is too long), line 78-80, among others throughout the text.
- Line 28. In materials and methods section, authors indicated that wtaer samples were collected in duplicate. Could you clarify that, please?
- Line 43. A reference must be included.
- Line 60. Are there any reported cases in Mexico from 2007?
- Line 206. Please, could authors explain what they mean with “water bodies”?
- Line 272. Why did you conducted the study only with samples collected in November? It would be interesting to perform these experiments in different seasons or including different months to compare (for expample October, November and December).
- I suggest revising the last part of the discussion. It would be convenient to finish this section in a different way.
- Line 357. Why did you point “colder months” if samples were all collected at the same time, in November?
Author Response
REVIEWER 3
Comments and Suggestions for Authors
The manuscript "Isolation and identification of Naegleria species in irrigation channels for recreational use in Mexicali Valley, Mexico" reports the presence of Naegleria in different areas in Mexicali Valley. This work is of interest in the field, as there are few reports about Naegleria infections and its incidence is understimated. Research design is appropriate and results are presented clearly. However, some changes must be done before being accepted to be published in Pathogens.
- - Manuscript elaboration. There are some grammar and punctuation mistakes that must be revised (for example: line 23, line 33, line 34, line 47, line 240, line 308 (line 308 is now 324): “…,it is capacity (now line 324)…”, etc.). Besides, some sentences may be rephrased to make them easier to understand. For example, lines 41-42, lines 44-47 (sentence is too long), line 78-80, among others throughout the text.
- All these changes were done along the document
- - Line 28. In materials and methods section, authors indicated that wtaer samples were collected in duplicate. Could you clarify that, please?
- There was a mistake in materials and methods section, now it was corrected as “triplicate” Lines 28 and 289
- - Line 43. A reference must be included.
- It was done, now line 42
- - Line 60. Are there any reported cases in Mexico from 2007?
- There are cases reported only in the local media. Unfortunately, they are not reported in the scientific literature. For example, officials with the Mexicali State Health Department (MSHD) reported the death of a 15-year-old boy from San Luis Rio Colorado Sonora in 2019 who had contracted the brain eating amoeba, Naegleria fowleri.
- - Line 206. Please, could authors explain what they mean with “water bodies”?
- “water bodies” means any natural or artificial aquatic environment such as lakes, rivers, or dams and irrigation canals. The word was changed by recreational waters in line 25, recreational aquatic sites (line 81); in line 206 by aquatic ecosystems, in lines 259-260 by aquatic environments
- - Line 272. Why did you conducted the study only with samples collected in November? It would be interesting to perform these experiments in different seasons or including different months to compare (for expample October, November and December).
- Now we are working with samples collected in Summer season
- - I suggest revising the last part of the discussion. It would be convenient to finish this section in a different way.
- The paragraph was edited. Lines 264 to 268
- - Line 357. Why did you point “colder months” if samples were all collected at the same time, in November?
- The words were changed by “in one of the coldest months of the year”. Now line 271
Submission Date31 July 2020
Date of this review19 Aug 2020 18:56:24

Round 2
Reviewer 1 Report
Dear Authors,
all requested changes have been made, resulting in a better quality manuscript that is easier to read.
However, I would like two minor changes before publication, we should mention the units used in line 88 (mg / L) "dissolved oxygen between 3.6 and 4.2", and specify what the ITS symbol (Internal transcribed spacer) corresponds to the term is first used in the manuscript line 148.
Best regards
Author Response
REVIEWER 1, ROUND 2
Comments and Suggestions for Authors
Dear Authors,
all requested changes have been made, resulting in a better quality manuscript that is easier to read.
However, I would like two minor changes before publication, we should mention the units used in line 88 (mg / L) "dissolved oxygen between 3.6 and 4.2",
- It was done, line 88
and specify what the ITS symbol (Internal transcribed spacer) corresponds to the term is first used in the manuscript line 148.
- it was done, lines 148 and 149
Best regards
Submission Date
31 July 2020
Date of this review
24 Sep 2020 13:01:19
Reviewer 2 Report
All my comments and considerations have been addressed. I strongly encourage authors to include in the final version the responses (results or discussion) provided to my comments.
Author Response
REVIEWER 2, ROUND 2
Comments and Suggestions for Authors
All my comments and considerations have been addressed. I strongly encourage authors to include in the final version the responses (results or discussion) provided to my comments.
- It was done in Discussion, lines from 257 to 276
This manuscript is a resubmission of an earlier submission. The following is a list of the peer review reports and author responses from that submission.
Round 1
Reviewer 1 Report
While I believe that the topic of the study has merit, I found the manuscript to be poorly written making it very difficult to read and interpret. The entire introduction, results and discussion need to be re-written to improve clarity and the use of language. In addition the sampling appears to be based on a single sample at each site and hence no replication. A minimum of 3 samples per site should be collected. The study did not include any other biological information on the sites, such as bacterial counts or bacterial species present, which have been show to impact the presence of N. fowleri, such as;
Goudot, S., Herbelin, P., Mathieu, L., Soreau, S., Banas, S. and Jorand, F. (2012) Growth dynamic of Naegleria fowleri in a microbial freshwater biofilm. Water Research 46(13), 3958-3966.
Morgan, M.J., Halstrom, S., Wylie, J.T., Walsh, T., Kaksonen, A.H., Sutton, D., Braun, K. and Puzon, G.J. (2016) Characterization of a Drinking Water Distribution Pipeline Terminally Colonized by Naegleria fowleri. Environmental Science & Technology 50(6), 2890-2898.
The study only used the physical chemical parameters to correlate to Naegleria's presence which is insufficient.
In addition the Flagellate test is not reliable to diagnose Naegleria species.
De Jonckheere, J. F., et al. (2001). "The amoeba-to-flagellate transformation test is not reliable for the diagnosis of the genus Naegleria. Description of three new Naegleria spp." Protist 152(2): 115-121.
And Line 194-197 are incorrect. Sifuentes et al. predominantly found more N. fowleri in winter and spring months compared to summer and fall.
Based on my assessment of the article I recommend it be rejected as the data is not enough and revisions are beyond the scope of what is provided.
Reviewer 2 Report
The present manuscript (Pathogens-802982) submitted by Patricia Bonilla-Lemus et al. “Isolation and identification of Naegleria species from irrigation channels used for recreational purposes in Mexicali Valley, México” have collected the Naegleria sample and morphologically and molecularly identified species and done the phylogenetic analysis with other Naegleria strains. This epidemiological less active amoeba species causes meningoencephalitis and mortality and hence it is significant study to categorize their pathogenicity to human or animals.
Apparently, here I am emphasizing some of the issues with this paper, which need to be answered reasonably and/or do the necessary amendments in the manuscript to make it convincing and reader friendly to the scientific community.
Here, one thing is surprising that how only using one pair of ITS primer, authors has molecularly identified different species of Naegleria. On the other hand, these samples were collected from the field, where having the high chances of gene alterations due to pollutants etc. Even though, the references cited by the authors, they themselves had also used species-specific amplifications. ITS1, 5.8s, ITS2 are the species-specific rDNA sequences. In general, for correct species identification researchers can not only rely on the nucleolar/nuclear gene. Most likely, they should also use mitochondrial genes like cytochrome oxidase (PMID: 27353585) to validate the accurate results. Or else, justify the primer designing strategy or cite any references to use only one pair of primer will identify the several species.
Did authors have submitted these ITS sequences of Naegleria species to NCBI? In table 3 they have exclusively presented the topmost findings of homology searches of BLAST results. But here they must include/provide the PCR amplified and sequenced product length with assigned NCBI ID, which would be self-sufficient. Relatively, figure 3 is not necessary to illustrate here, because there are many numbers of sequences and gaps in it, and not much informative.
How many Naegleria cells did you find in the collection of 250 ml of water sample. This would also be helpful and good informative signal for the hazardous index for irrigation system of that locality.
Physicochemical parameters like water temperature, conductivity, pH and dissolved oxygen, make significance relevance with the occurrence of this amoeba, in a specific habitat. It would be a concrete information, if table 1 and 2 merged with only relevant and necessary parameters. Use the units of conductivity parameters wherever its necessary (Ex- Line No. 206).
Author should prevent themselves to use unnecessary acronyms which is not needed or been used very minimal number of times like FLA, NNA, DO. Use the word dissolved oxygen in Table 1.
Although author/s has tried to do some good work but there are few issues which demanding their logical answers and ultimately need to improvise the manuscript. However, I have pointed out here, few of them and assuming that author will take of rest including English and grammar.
I hope author will consider all the suggestions and follow them. All the Best.